# The impact of university students' computational thinking on AI literacy: A longitudinal study based on SEM-PLS

Qiuyu Li[1], Tinglan Huang[2], Lehui Huang[1]*, Xueshi Wu[1]

1 Faculty of Education, Jiangxi Science and Technology Normal University, Nanchang, Jiangxi, China,
2 Educational Leadership Program, Emilio Aguinaldo College, Manila, Philippines

* hlh8899@jxstnu.edu.cn

## Abstract

Artificial intelligence literacy is a core competency that underpins individual development. Although previous studies have demonstrated a cross-sectional association between computational thinking and artificial intelligence literacy, there has been little analysis of their longitudinal effects. This study utilized a sample of 216 undergraduate students from southern China and employed structural equation modeling (SEM-PLS) and paired-sample t-tests. Through a two-stage longitudinal data analysis, the study found that computational thinking has a significant positive effect on artificial intelligence literacy and that both competencies improved significantly over time. By tracking the evolution of these two competencies, this study offers a novel longitudinal perspective, confirms their synchronous development, and highlights the stability of their cross-sectional relationship, thereby providing theoretical foundations and methodological references for educational practice.

## Introduction

Recent breakthroughs in generative artificial intelligence (GenAI) technologies—exemplified by advanced systems such as DeepSeek—have triggered a cognitive shift in education, fundamentally reshaping the underlying logic of knowledge transmission [1]. This deep integration of technological innovation and educational paradigms has given rise to the pivotal concept of artificial intelligence literacy (AIL). Long and Magerko [2] defined AIL as "a set of competencies that enable individuals to critically evaluate AI technologies, communicate and collaborate effectively with AI, and utilize AI as a tool across online, domestic, and workplace contexts." This literacy is widely recognized as comprising four dimensions: awareness, application, evaluation, and ethics [3]. A recent study published in Nature Methods further underscores the growing importance of developing AI literacy [4]. In response, leading academic institutions such as Stanford University and the University of Oxford have begun formulating policies governing the use of generative AI tools—including ChatGPT, Bard,

**Data availability statement:** All relevant data are within the paper and its Supporting information files.

**Funding:** Study Title: Research on the Development of VR Biology Teaching Resources for Junior High Schools in the Context of Educational Digital Transformation Author Initials: H Grant Number: 23YB139 Funder Name: Jiangxi Provincial Department of Education Funder URL: http://jyt.jiangxi. gov.cn Funder Role: Funder provided financial support during the preparation ofthe manuscript Study Title: Digital Evaluation of Digital Transformation of Vocational Education: Research on the Construction of Maturity Model and Strategy Optimization Author Initials: H Grant Number: 2024XJZD004 Funder Name: Jiangxi Science and Technology Normal University Funder URL: http://www.jxstnu.edu. cn Funder Role: Funder provided financial support during the preparation ofthe manuscript Study Title: Research on the Practical Path of AI-assisted Classroom Learning Situation Analysis for College Students Author Initials: W Grant Number: 2024XJZD003 Funder Name: Jiangxi Science and Technology Normal University Funder URL: http://www.jxstnu. edu.cn Funder Role: Funder provided financial support during the preparation ofthe manuscript Study Title: A study on digital transform ation pathways for new quality productivity in wocational education Author Initials: Z Grant Number: YC2024-S717 Funder Name: Jiangxi Provincial Department of Education Funder URL: http://jyt.jiangxi.gov.cn Funder Role: Funder provided financial support during the preparation ofthe manuscript.

**Competing interests:** The authors have declared that no competing interests exist.

DALL·E, and Stable Diffusion—emphasizing their ethical and appropriate use within academic settings. At the international level, the United Nations Educational, Scientific, and Cultural Organization (UNESCO) formally established a global educational framework for artificial intelligence literacy in its 2019 Beijing Consensus on Artificial Intelligence and Education. Building on this framework, China has actively implemented corresponding initiatives. In 2021, the China Educational Technology Center issued the *Framework for Students' AI Technology and Engineering Literacy*, an official standard that delineates a matrix of core competencies students are expected to develop to adapt to the demands of the age of artificial intelligence. In June 2024, Zhejiang University released the *Red Paper on University Students' AI Literacy (2024 Edition)*, which serves as an academic reference for AI literacy in higher education. Subsequently, between October and November, Beijing and Chongqing issued their respective *Guidelines for the Application of Artificial Intelligence in Education*. These documents not only establish a three-dimensional competency framework encompassing "standards–scenarios–skills," but also emphasize the development of general education courses in generative AI. Taken together, these initiatives aim to systematically enhance students' adaptability and creativity in an increasingly intelligent and digital society [5]. The formulation of these policies is not coincidental: empirical evidence from Ng, Maduranayagam [6] indicates a significant gap in the appropriate and normative use of AI tools in academia. Enhancing AI literacy thus goes beyond a purely technical understanding of artificial intelligence; it also entails a critical awareness of the ethical and societal implications of its application.

Scholarly research on AI literacy has predominantly targeted primary and secondary school teachers and students, as well as undergraduates. For instance, Ng, Su [7] conducted a comprehensive review of AI literacy education in secondary schools, identifying commonly used instructional methods, tools, content, assessment strategies, and measurement dimensions. As target populations have been more precisely defined, the academic community has gradually developed a paradigm for measuring AI literacy and identifying its influencing factors. Methodologically, scholars have primarily employed structural equation modeling, individual interviews, and survey questionnaires. For example, Du, Sun [8] adopted a two-step structural equation modeling (SEM) approach, revealing that K–12 teachers' perceptions of AI for social good and their self-efficacy in learning AI were two direct determinants of their behavioral intention to engage in AI learning. Although such quantitative studies have identified direct determinants of behavioral intention, systematic exploration of the underlying mechanisms that shape AI literacy remains limited.

Computational thinking is an essential cognitive skill for university students [9]. It refers to the application of core principles and fundamental concepts from computer science to problem solving and system design, encompassing a range of cognitive activities that involve conceptualizing and addressing problems in the manner of a computer scientist [10]. Korkmaz, Çakir [11] proposed that computational thinking skills comprise five dimensions: algorithmic thinking, collaboration, creativity, critical thinking, and problem solving. Individuals with strong computational thinking skills are more likely to be proficient in understanding and using AI-related concepts [12]. In a

comprehensive literature review, Celik [13] highlighted gaps in AI literacy research, particularly the limited exploration of its influencing factors. Specifically, based on empirical analysis, the author proposed a three-dimensional analytical framework encompassing the digital divide, computational thinking, and cognitive absorption, and empirically confirmed the significant positive impact of computational thinking on AI literacy. However, this study was limited to validating correlations between variables and did not examine the magnitude or dynamic evolution of the effects of computational thinking on AI literacy. In terms of research paradigms, existing studies on AI literacy are predominantly cross-sectional, with relatively few longitudinal investigations. Empirical research examining the dynamic influence of variables such as computational thinking on AI literacy remains particularly scarce.

This study aims to investigate the impact of computational thinking on the development of AI literacy over time. To achieve this aim, a dynamic partial least squares structural equation model (PLS-SEM) was constructed based on cross-sectional and longitudinal data to systematically analyze the relationship between computational thinking and AI literacy over time. Accordingly, this study addresses the following research questions:

a) Does computational thinking have a significant impact on AI literacy?

b) Does the influence of computational thinking on AI literacy change over time?

c) Does computational thinking at an earlier time point predict AI literacy at a subsequent time point?

This study features two core innovations. First, it closely links computational thinking with artificial intelligence literacy, analyzing computational thinking as a key variable to determine its impact on AI literacy. Second, building upon a detailed cross-sectional data analysis, the study further collected two months of longitudinal data to reveal how the state of computational thinking at the initial time point (T0) influences the level of artificial intelligence literacy at the subsequent time point (T1).

## Literature review

The concept of AI literacy was initially proposed by Kandlhofer, Steinbauer [14], referring to the ability to understand and apply the technologies and concepts underlying artificial intelligence and computer science. This includes knowledge of areas such as graphs and data structures, sorting and searching algorithms, problem-solving, and machine learning. Long and Magerko [2] further defined AI literacy as "a set of competencies that enable individuals to critically evaluate AI technologies, communicate and collaborate effectively with AI, and utilize AI as a tool across online, domestic, and workplace contexts." Wang, Rau [3] expanded this construct by proposing a widely recognized framework consisting of four dimensions: awareness, application, evaluation, and ethics.

Existing research has primarily focused on K–12 teachers and students, as well as undergraduate populations. For instance, Kong, Man-Yin Cheung [15] conducted a study with university students in Hong Kong and found that short-term courses significantly enhanced their AI literacy and sense of empowerment while reducing disciplinary and gender disparities. In a systematic review of AI literacy education in secondary schools, Ng, Su [7] summarized prevalent instructional methods, tools, and assessment strategies. Nevertheless, current studies remain largely confined to specific educational stages, with a noticeable lack of cross-sectional and cross-cultural comparative analyses. Furthermore, longitudinal research investigating the developmental mechanisms of AI literacy remains limited.

Computational thinking (CT) is defined as a set of cognitive activities that involve problem-solving and system design using fundamental computer science concepts [10]. A framework proposed by Korkmaz, Çakir [11], comprising five dimensions—algorithmic thinking, collaboration, creativity, critical thinking, and problem solving—has been widely adopted. Empirical studies indicate that learning programming languages such as Scratch significantly enhances CT skills among primary school students [16]. Additionally, factors such as gender stereotypes, learning motivation, and collaborative learning have been shown to substantially influence the development of CT [17,18]. Research participants have ranged

from preschool children to pre-service teachers. For example, a longitudinal study by Zhang and Wong [19] on primary school students identified gender, prior computer experience, and formal learning opportunities as significant predictors of CT competency. Similarly, Sun, You [20] found that various background factors influence the computational thinking abilities of STEAM teachers in China. Although diverse methodologies such as SEM, meta-analysis, and quasi-experimental designs have been employed [21,22], most studies remain cross-sectional, making it difficult to infer causal pathways or long-term developmental trajectories of CT.

Methodologically, studies on AI literacy often adopt mixed-method designs that incorporate structural equation modeling, interviews, questionnaires, and focus groups. For instance, Zhao, Wu [23] used SEM to elucidate the structural relationships among multiple dimensions of AI literacy. Du, Sun [8] identified teachers' self-efficacy and perceived social utility as key factors influencing their willingness to engage in AI learning. Through qualitative analysis, Velander and Taiye [24] highlighted that the often incidental nature of teachers' AI knowledge acquisition can lead to misunderstandings. Despite these methodological advances, there is a notable scarcity of longitudinal studies that jointly examine AI literacy and computational thinking in an integrated manner. The interaction between these two constructs, therefore, remains poorly understood and warrants further empirical investigation.

## Hypothesis

Computational thinking (CT) refers to the cognitive processes involved in formulating problems and designing solutions through a series of structured steps [25]. It represents a universally applicable set of attitudes and skills [10]. Competence in problem solving—a core component of CT—is essential across disciplines, including computer science and engineering, and is increasingly critical in diverse domains of human activity [26]. Furthermore, CT encompasses problem solving, critical thinking, and decision making, abilities that are indispensable for navigating the complex and dynamic landscape of artificial intelligence [26]. As such, CT provides a foundational framework for reasoning that enhances both the understanding and the development of AI systems. Empirical evidence from Lin, Zhou [27] indicates that CT significantly predicts AI literacy among university students. Celik [13] examined factors influencing AI literacy among university students and found a positive correlation between students' CT and their level of AI literacy. Greenwald, Leitner [28] emphasized the importance of leveraging concepts such as CT to help students better understand AI-related knowledge and address AI-related challenges. Similarly, Kim, Jang [29] identified CT as essential for acquiring AI literacy. Based on these findings, the present study hypothesizes that the stronger university students' CT skills are, the more aware, evaluative, and ethically reflective they will be when engaging with AI technologies. Accordingly, this study proposes the following hypotheses:

H1: At Time 0 (T0), university students' CT has a significant positive effect on their AI literacy.

H2: At Time 1 (T1), university students' CT has a significant positive effect on their AI literacy.

H3: CT at Time 0 (T0) significantly and positively predicts AI literacy at Time 1 (T1) among university students.

Kandlhofer, Steinbauer [14] first introduced the concept of artificial intelligence literacy, which involves understanding and applying core technologies and principles in AI and computer science. It encompasses knowledge across various domains, including graph and data structures, sorting algorithms, search-based problem solving, classical planning methods, logic, and machine learning. Lei, Fu [30] demonstrated that university students' computational thinking can be significantly enhanced when instructors implement flexible group-based instruction, guide students through in-depth negotiation, and provide timely feedback. Similarly, Sun, Hu [31] found that computational thinking among university students is positively associated with both individual factors (e.g., gender, academic major) and contextual factors (e.g., educational environment, institutional support). Moreover, Ng, Chen [32] showed that gamified approaches can provide emotional and cognitive support to learners and are positively associated with AI literacy. Based on these findings, this study posits that

multiple factors influence computational thinking and AI literacy and represent dynamic, evolving constructs. Accordingly, the following hypotheses are proposed:

H4: University students' computational thinking differs significantly between Time 0 (T0) and Time 1 (T1).

H5: University students' AI literacy differs significantly between Time 0 (T0) and Time 1 (T1).

Fig 1 presents the proposed research model based on the above hypotheses.

## Methods

### Participants

The two-wave survey was administered via the Questionnaire Star online platform on 10 September and 10 November 2024, with a two-month interval between the two measurement points. The length of this interval was determined on the basis of methodological recommendations by Wang and Zhang [33], who suggested using half of the peak time for indirect effects, and is further supported by the recommendations of Wu, Li [34]. Although a longer interval might better capture long-term change, Zhao, Shipp [35] demonstrated an inverted U-shaped relationship between time lag and observable effect sizes. Therefore, the two-month interval adopted in this study is expected to fall within the ascending phase of this curve, thereby enabling more sensitive detection of the dynamic relationship between computational thinking and AI literacy.

This study was conducted with students from a public university in southern China and employed a stratified sampling method for participant selection. Students were first categorized by academic year in the Learning Management System (LMS) and then ordered by student identification number within each year. A systematic sampling procedure was applied, selecting one student for every 100 individuals. A total of 301 students consented to participate and were recruited via email invitations. Prior to completing the questionnaire, participants received detailed information about the study, including its objectives and measures for protecting privacy. The survey was administered only after participants completed an electronic, signature-based informed consent form, and all participants retained the right to withdraw from the study at any time without penalty. The study was conducted in accordance with relevant guidelines and regulations governing research involving human participants (e.g., the Declaration of Helsinki or equivalent ethical standards). The study protocol was submitted by Jiangxi Science and Technology Normal University to the Jiangxi Provincial Department of Education and was approved under authorization number 23YB139.

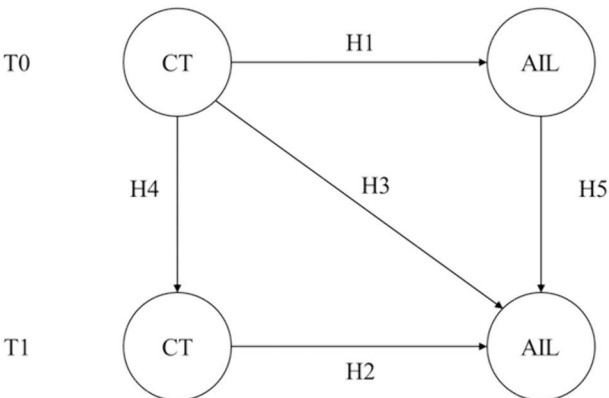

**Fig 1. Hypothesized conceptual framework.**

After the necessary preparations were completed, data were collected from 301 undergraduate students at the university. To ensure sample quality, several screening criteria were applied, following established methodologies in the literature [36–39]. The time taken to complete the questionnaire served as one screening benchmark, whereas uniform response patterns (e.g., identical answers to all items) were treated as indicative of inattention or careless responding [37,39,40]. Based on these references, the following criteria were used to identify valid responses: (1) The questionnaire was designed to take approximately 2–5 minutes to complete; participants who completed it in less than 90 seconds were considered to have responded too rapidly, and their data were excluded. (2) Questionnaires showing uniform response patterns (e.g., all "1" or all "5") or other extreme patterns were also excluded. (3) All IP addresses were recorded and checked for duplication; repeated entries were removed. After the initial screening, 12 invalid responses were removed, resulting in 289 valid questionnaires. In the second wave of data collection, 39 participants from the first wave did not respond, and an additional 34 responses were deemed invalid. Consequently, 216 valid responses remained for further analysis (S1 Data). To assess whether attrition introduced selective bias, Mann-Whitney U tests and chi-square tests of homogeneity were conducted to compare completers and non-completers on baseline measures. The results were consistent with the t-test findings (CT score: $U = 8082.500$, $p = 0.105$; AIL score: $U = 8517.500$, $p = 0.329$). Furthermore, the chi-square test indicated no significant difference in gender distribution between completers and non-completers ($\chi^2(1) = 0.856$, $p = 0.355$). These results suggest that attrition was random rather than systematic. The missing data mechanism is therefore consistent with a Missing Completely at Random (MCAR) pattern, posing minimal threat to the study's internal validity.

Hair, Risher [41] suggested that the minimum sample size for structural equation modeling typically ranges from 100 to 150, even for relatively simple models. For more complex models—for example, those involving multiple latent variables, numerous paths, or many free parameters—a sample size of 200–400 or more is recommended. Therefore, the sample size of 216 in this study meets the recommended standards for SEM analysis.

Among the 216 valid responses, 84 were from male participants and 132 from female participants. A multi-group analysis was conducted to examine gender invariance across key model paths (T0 CT→T0 AIL: two-tailed $p = 0.313$; T0 CT→T1 AIL: two-tailed $p = 0.483$; T1 CT→T1 AIL: two-tailed $p = 0.534$). The results indicate that the core model is invariant across gender groups, supporting the structural stability of the hypothesized relationships and the representativeness of the sample. By academic year, the sample consisted of 55 freshmen, 56 sophomores, 89 juniors, and 17 seniors. Detailed demographic characteristics, including gender distribution, academic year, and field of study, are summarized in Table 1.

## Measurement instruments

AI literacy was measured using a scale adapted from Laupichler, Aster [42], which was specifically developed to assess university students' AI literacy. The scale consists of 38 items (e.g., "I can identify ethical issues surrounding artificial intelligence") and has been empirically validated in subsequent research by Laupichler, Aster [43] (S1 Table).

Table 1. Basic demographic information of participants.

|  | Category | Quantity | Percentage |
|---|---|---|---|
| **Gender** | Male | 84 | 39% |
|  | Female | 132 | 61% |
| **Grade** | Freshman | 55 | 25% |
|  | Sophomore | 56 | 26% |
|  | Junior | 89 | 41% |
|  | Senior | 17 | 8% |
| **Subject** | Natural Sciences | 70 | 32% |
|  | Humanities and Social Sciences | 146 | 68% |

 

Computational thinking was assessed using a scale adapted from Korkmaz, Çakir [11], which was designed to measure CT among university students. The scale consists of 14 items (e.g., "When faced with a problem, I can quickly construct a mathematical model to solve it"). The scale demonstrated good internal consistency, with a Cronbach's alpha of 0.822.

## Statistical analysis

Partial least squares structural equation modeling offers several advantages for data analysis. It can yield valid results with relatively small sample sizes and does not require the data to follow a normal distribution [41]. PLS-SEM is particularly effective for handling complex structural models with multiple dimensions and can simultaneously estimate both reflective and formative indicators. These characteristics make PLS-SEM a powerful tool for researchers, providing a comprehensive perspective on the relationships among variables. As an exploratory approach, PLS-SEM emphasizes theory development rather than theory confirmation. It is also well-suited for prediction and can yield accurate predictive results [41]. SmartPLS 4.0 was used to analyze the relationships among variables within the research model. The Smart-PLS analysis included an assessment of the measurement model (outer model), in which the constructs' validity and reliability were evaluated. In the structural model (inner model), Hypotheses 1, 2, and 3 were then tested. Additionally, paired-samples t-tests were conducted in SPSS to examine changes in computational thinking and AI literacy between the two time points.

## Results

### Measurement model

The measurement model in this study was evaluated using PLS-SEM to assess the reliability and validity of the latent constructs. Composite reliability (CR) reflects the reliability of a construct, whereas Cronbach's alpha provides a lower-bound estimate of that reliability [41]. According to Fornell and Larcker (44), the measurement model is considered adequate in reliability when both CR and Cronbach's alpha exceed 0.70. At Time 0 (T0), the composite reliability (CR) values of the constructs were 0.950 and 0.898, and at Time 1 (T1), the CR values were 0.947 and 0.831 (see Table 2). The Cronbach's alpha values at T0 were 0.947 and 0.884, whereas at T1 they were 0.942 and 0.820 (Table 2). These results indicate that all constructs demonstrated good internal consistency at both time points, confirming that the measurement model exhibits its adequate reliability [41].

Validity refers to the extent to which the scale items accurately measure the intended constructs and is typically assessed through convergent and discriminant validity. Convergent validity reflects the degree of intercorrelation among items within the same construct [41]. It is typically evaluated using factor loadings and each construct's average variance extracted (AVE), with AVE values greater than 0.50 indicating acceptable convergent validity [44]. At Time 0 (T0), the AVE values for the constructs were 0.545 and 0.632, and at Time 1 (T1), they were 0.522 and 0.527, all exceeding the

Table 2. Composite reliability (CR).

|  | Time | Cronbach's alpha | CR | AVE |
|---|---|---|---|---|
| **AIL** | T0 | 0.947 | 0.950 | 0.545 |
|  | T1 | 0.942 | 0.947 | 0.522 |
| **CT** | T0 | 0.884 | 0.898 | 0.632 |
|  | T1 | 0.820 | 0.831 | 0.527 |

recommended threshold of 0.50. According to Fornell and Larcker (44), these results indicate that the model exhibits adequate convergent validity (Table 2).

Discriminant validity refers to the extent to which each construct in the model is empirically distinct from the others. It is commonly assessed using the Fornell–Larcker criterion and the heterotrait–monotrait ratio of correlations (HTMT) [41]. According to the Fornell–Larcker criterion, the square root of each construct's AVE should be greater than its correlations with other constructs, and the AVE should exceed 0.50. The AVE correlation matrix and the square roots of AVE for computational thinking and AI literacy are presented in Table 3. The results indicated that the two variables were significantly correlated at both T0 and T1 (p<0.001). The square roots of the AVE values for each construct at T0 were 0.738 and 0.795, and at T1 were 0.723 and 0.726. All of these values exceeded the recommended threshold of 0.50 and were greater than the Pearson correlation coefficients between the two variables, thus satisfying the Fornell–Larcker criterion. HTMT analysis was also conducted to assess whether the constructs could be empirically distinguished. Hair, Risher [41] state that an HTMT value below 0.85 indicates adequate discriminant validity between two constructs. All HTMT values in this study—0.537, 0.617, and 0.092—were well below this threshold (Table 4), confirming that the model exhibits strong discriminant validity.

## Structural model

The structural model was assessed for collinearity using variance inflation factors (VIFs) [41]. A VIF value below 5 indicates acceptable collinearity and suggests that no critical multicollinearity issues are present in the model [41]. In this study, the VIF values were 1.000, 1.019, and 1.019, all well below this threshold, indicating that the model meets recommended standards regarding collinearity (Table 5).

**Table 3. Discriminant validity (Fornell-Larcker criterion).**

|  | AIL (T0) | AIL (T1) | CT (T0) | CT (T1) |
|---|---|---|---|---|
| **AIL（T0）** | **0.738** |  |  |  |
| **AIL（T1）** | −0.001 | **0.723** |  |  |
| **CT（T0）** | 0.507 | −0.056 | **0.795** |  |
| **CT（T1）** | −0.038 | 0.566 | −0.136 | **0.726** |

**Table 4. Discriminant validity (HTMT).**

|  |  | Heterotrait-monotrait ratio (HTMT) |
|---|---|---|
| T0 | CT<->AIL | 0.537 |
| T1 | CT<->AIL | 0.617 |
| T0-T1 | CT<->AIL | 0.092 |

**Table 5. Formative indicator loadings.**

|  |  | VIF |
|---|---|---|
| T0 | CT<->AIL | 1.000 |
| T1 | CT<->AIL | 1.019 |
| T0-T1 | CT<->AIL | 1.019 |

 

PLS-SEM employs bootstrapping—a resampling technique—to estimate the significance of the path coefficients in the structural model. In this study, 5,000 bootstrap samples were generated to compute the path coefficients (β) and their corresponding t-values [41]. The standardized path coefficient (β) represents the strength and direction of a regression path. In contrast, the t value is used to test the statistical significance of relationships in both the measurement (outer) and structural (inner) models. A path coefficient close to 0.5 in both the original sample and the bootstrapped mean estimates generally indicates a moderate-to-strong effect [41]. The results from the T0 dataset indicate that computational thinking significantly and positively predicts AI literacy (β = 0.507, t = 9.682, p < 0.001). Similarly, the T1 data demonstrate a significant positive relationship between computational thinking and AI literacy (β = 0.569, t = 11.112, p < 0.001). However, computational thinking at T0 did not significantly predict AI literacy at T1 (β = 0.021, t = 0.375, p = 0.708; Table 6). The structural model results are illustrated in Fig 2.

### Significance testing of computational thinking and AI literacy across time points

The t-test (Student's t-test) is suitable for normally distributed data with small sample sizes (n < 30) and unknown population standard deviations. It is used to assess the difference between two means. The Shapiro-Wilk test indicates that the difference scores for CT (W = 0.988, p = 0.056) and AIL (W = 0.996, p = 0.910) are normally distributed. This study employed a paired-samples t-test to analyze levels of computational thinking and artificial intelligence literacy across time points. The calculations yielded a t-value of 7.241 for the computational thinking group (p < 0.001) and a t-value of 5.867 for the artificial intelligence literature group (p < 0.001). In summary, Hypotheses 4 and 5 are supported, indicating that compared to time point T0, there were significant changes in both computational thinking and artificial intelligence literacy among college students at time point T1 (see Table 7).

**Table 6. Path coefficients.**

|  | β | T statistics | [2.5%,97.5%] | P | Result |
|---|---|---|---|---|---|
| **H1: CT -> AIL** | 0.507 | 9.682 | [0.412-0.617] | 0.000 | H1 supported |
| **H2: CT -> AIL** | 0.569 | 11.112 | [0.471-0.673] | 0.000 | H2 supported |
| **H3: CT -> AIL** | 0.021 | 0.375 | [-0.089-0.139] | 0.708 | H3 reject |

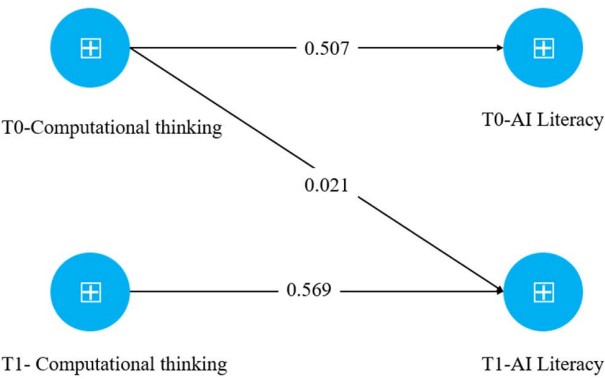

**Fig 2. Results of the research model.**

**Table 7. Paired samples T-Test.**

|  | T statistics | P |
|---|---|---|
| **CT** | 7.241 | .000 |
| **AIL** | 5.867 | .000 |

## Discussion

### Discussion of findings

**Discussion of structural model paths.** The primary aim of this study was to examine the influence of computational thinking on AI literacy from both cross-sectional and longitudinal perspectives. A total of 216 valid responses were collected using the computational thinking scale (CTS) and the AI literacy scale, with two rounds of data collection conducted via the Questionnaire Star online platform in September and November 2024. The results indicate that computational thinking has a significant positive impact on AI literacy across the study period. The research confirms that both students' computational thinking and AI literacy improved significantly over time.

Analyses conducted using SmartPLS 4.0 confirmed Hypotheses 1 and 2, demonstrating a statistically significant positive influence of computational thinking on AI literacy in the cross-sectional models. This finding aligns with one of the key conclusions of Celik [13], who identified computational thinking as a significant determinant of AI literacy in his proposed model. Computational thinking was shown to facilitate the use, identification, and evaluation of AI-based technologies. Our results suggest that computational thinking should be regarded as a critical component in the development of AI literacy and highlight the importance of incorporating targeted computational thinking interventions into educational practice to enhance AI literacy.

A noteworthy finding of this study is that, while cross-sectional analyses revealed a significant positive association between CT and AIL (supporting H1 and H2), longitudinal analyses showed that CT at Time 0 (T0) did not significantly predict AIL at Time 1 (T1), thus failing to support H3. This result highlights the complexity of the relationship between CT and AIL and warrants in-depth discussion. First, cross-sectional relationships reflect covariation between skills at a single point in time, whereas longitudinal relationships test predictive validity. The cross-sectional findings indicate that students with stronger CT skills at a given point in time also tend to possess higher levels of AIL. This covariation may stem from project-based learning or problem-solving tasks, in which cognitive processes inherent in CT—such as abstraction, decomposition, and algorithmic reasoning—are simultaneously activated and reinforced alongside the competencies required to understand and apply AI technologies, thereby creating a synergistic effect [45,46]. However, during the two-month instructional period in this study, the development of students' AIL was likely influenced not only by CT but also by multiple other factors. These factors include the progressive delivery of course content, diverse project-based practices, peer interactions during collaborative activities, and targeted guidance from instructors [47,48]. These rich learning experiences may have diluted or obscured the independent predictive power of CT over time, resulting in its non-significant longitudinal effect. This interpretation aligns with existing research suggesting that the transfer effects of CT may depend on specific learning contexts and task supports [49]. Second, potential ceiling effects or issues with measurement sensitivity may also have played a role. The relatively high baseline CT scores at T0 may have limited the scope for further CT-based predictive gains in AIL. Moreover, although the AIL assessment tool demonstrated good reliability and validity in cross-sectional contexts, it may lack sufficient sensitivity to detect subtle changes longitudinally—especially in the absence of more differentiated task-based scenarios [50] and given the relatively short duration of the instructional period [51]. In conclusion, the relationship between CT and AIL cannot be fully captured by a simple linear predictive model. The cross-sectional results emphasize the close cognitive connection between the two constructs. In contrast, the longitudinal findings suggest that, in authentic educational environments, the development of AIL is moderated by multiple factors, including instructional context, student background, and curriculum design [52,53].

The results of the paired-samples t-test confirmed Hypothesis 4, indicating a significant change in university students' CT between Time 0 (T0) and Time 1 (T1). During the instructional period, factors such as students' interest in coding and prior coding experience can significantly influence the development of CT, leading to notable variation over time [19]. Furthermore, when instructors adopt flexible strategies such as shared regulation and group-based learning, students' CT can be further enhanced [30]. These findings are consistent with the results of the present study, which demonstrate that CT undergoes significant positive change over time.

Regarding Hypothesis 5, the results indicated that university students' AIL significantly improved from Time 0 (T0) to Time 1 (T1). This finding is consistent with the results of Moreno-León and Vasco-González [54], who examined students in Spanish schools and explored how programming activities affect CT and AIL. Their study showed that using platforms such as Scratch and Machine Learning for Kids significantly enhanced students' CT skills and AIL. Similarly, the present findings align with those of Ng, Su [7], who demonstrated that, when properly implemented, gamified learning approaches can provide emotional and cognitive support, thereby promoting students' AIL.

**Discussion of potential negative effects of AI in education.** Although AI literacy offers significant benefits for learning and skill development, we must also recognize the potential negative impacts and risks associated with integrating AI into educational settings. Overreliance on AI tools may undermine critical thinking and problem-solving skills, as students may shift cognitive effort onto AI systems rather than engaging in deep reasoning [55–57]. Reliance on AI, particularly among learners with weaker self-regulation skills, may lead to reduced academic autonomy, a loss of decision-making capacity, diminished initiative, and weakened innovation capabilities [55,58–60]. Furthermore, ethical issues surrounding AI—including data privacy, informed consent, and the potential risks of academic misconduct—further complicate the application of AI in education [6,61]. Therefore, fostering AI literacy must go beyond technical proficiency to include a critical understanding of these risks. Educators and policymakers should integrate discussions on AI ethics, bias mitigation, and the responsible use of AI into the curriculum to ensure that students are not only able to use AI effectively but are also equipped to question and critically evaluate its applications.

**Discussion of external factors.** This study, while examining the relationship between AIL and CT, did not explicitly account for the potential influence of external moderating variables such as gender, prior AI experience, and academic achievement. The role of gender warrants particular attention, as existing research indicates that gender differences may play a significant role in the development and manifestation of both AIL and CT. For instance, Kong, Man-Yin Cheung [15] found that short-term AI courses significantly reduced disparities in AIL and self-efficacy across genders and academic backgrounds, suggesting that gender may serve as an important moderating variable. Similarly, Master, Tang [17] highlighted that student motivation and gender stereotypes significantly influence the development of CT, whereas Lai, Ye [18] further suggested that collaborative learning can help mitigate gaps related to gender and prior experience, thereby promoting more equitable learning outcomes. Together, these findings imply that gender and other background factors may indirectly moderate the relationship between AIL and CT by influencing learning motivation, engagement, and access to resources. Therefore, future research should more systematically incorporate these variables to develop a more comprehensive understanding of the mechanisms through which AIL and CT interact.

## Practical implications

This study provides strong evidence that computational thinking significantly and positively influences AI literacy, thereby establishing its role as a core determinant of AI literacy. This finding offers valuable guidance for educators seeking to cultivate AI literacy among students. Specifically, it suggests that efforts should focus on three key dimensions: systematic instructional design, upgrading paradigms for learning environments, and fostering innovation in technology-based practices.

In designing and implementing AI-related courses, educators are encouraged to incorporate structured CT training into instruction systematically. This integration can be achieved through carefully designed modular programming tasks,

such as screen-free activities, that enhance students' hands-on skills. Additionally, heterogeneous group collaboration structures can promote peer interaction and cooperation, thereby strengthening key CT skills such as problem decomposition, pattern recognition, and algorithm design. To ensure effective learning outcomes, instructors can adopt a "dual-cycle" instructional model, in which explicit CT training is integrated concurrently with programming practice. For example, during a project on image recognition, instructors can guide students to use flowcharts to deconstruct the complexities of machine learning processes. This approach helps students build cognitive bridges between CT and AI literacy, enabling them to deepen their understanding of both through hands-on, iterative practice.

Establishing a Diversified AI Ecosystem in Higher Education. Universities should develop a comprehensive, multidimensional support system to cultivate AI literacy, grounded in both physical space and cultural context. At the physical level, higher education institutions should implement targeted strategies to enhance education through digital and intelligent technologies, tailored to their institutional missions and disciplinary strengths. This involves the creation of a "virtual–physical symbiotic" intelligent education ecosystem. Such an ecosystem encompasses the development of physical teaching and research spaces equipped with intelligent interactive devices, data analytics platforms, and other advanced infrastructure. It also includes the construction of digital-twin virtual environments that support immersive learning and collaborative research. By integrating virtual and physical resources, this environment can provide intelligent teaching laboratories and innovative practice platforms for faculty and students. At the cultural level, universities should deepen the integration of industry and education by collaborating with enterprise partners to co-create an innovation-oriented culture centered on human–AI collaboration. By constructing a synergistic mechanism that integrates the "education chain–industry chain–innovation chain," institutions can promote the formation of interdisciplinary research teams and cross-cultural academic exchange platforms. This approach aims to foster an open, inclusive culture of innovation that encourages faculty and students to explore research paradigm shifts driven by digital and intelligent technologies. It also supports the development of human–AI collaborative research methods, thereby forming a virtuous cycle of technological development, application validation, and theoretical innovation. Together, the dual driving forces of physical space and cultural context act to enhance the AI literacy of faculty and students in higher education institutions.

Developing a national-level personalized AI learning support system is a strategic priority. At this level, efforts should focus on building an intelligent, AI-powered learning diagnostics platform, supported by advanced learning analytics, to address students' personalized learning needs and enhance AI literacy. Such a platform can leverage multimodal data—including programming logs, eye-tracking records, and concept maps—to construct fine-grained cognitive profiles of learners. Based on these profiles, educators can implement dynamic grouping strategies grounded in the Zone of Proximal Development (ZPD) to customize learning trajectories for individual students. For example, students who exhibit weaker abstract thinking skills may benefit from intuitive and accessible machine learning visualization tools to scaffold their understanding. In contrast, students with strong algorithmic optimization abilities can be engaged in open-ended innovation projects designed to stimulate their creative potential.

### Research limitations and future directions

Regarding sample size, although the final sample of 216 participants met the minimum requirements for structural equation modeling, the sample size remains relatively small, which may limit the generalizability of the findings. To ensure that the results were not significantly affected by sampling bias, we employed stratified and systematic sampling procedures. However, the sample in this study was drawn exclusively from a single university in southern China, and the current gender ratio is imbalanced, with a male-to-female ratio of 2:3. This may limit the applicability of the findings to a broader population. Future research should employ larger, more diverse samples drawn from multiple institutions and different regions to enhance the external validity of the findings.

It is particularly worth noting that the data for this study were collected in 2024, at a time when the field of artificial intelligence is undergoing rapid evolution. While the fundamental relationship between computational thinking and AI literacy

may remain stable, the specific manifestations and application scenarios of AI literacy may have changed since data collection was conducted. The core findings regarding the association between computational thinking and AI literacy remain relevant, as they reflect underlying cognitive processes that are less susceptible to rapid technological change. Nevertheless, readers should interpret the results in the context of the rapidly evolving landscape of AI development.

When assessing the level of artificial intelligence literacy, this study modeled computational thinking as the sole outcome variable and did not incorporate other potentially important influencing factors—such as indicators of the digital divide—which limits the comprehensiveness of the findings. Furthermore, both computational thinking and AI literacy were operationalized as higher-order composite constructs, without examining the nuanced relationships among their specific sub-dimensions (e.g., algorithmic thinking and critical thinking within computational thinking; awareness, evaluation, and ethical reasoning within AI literacy). Although this approach facilitates macro-level analysis, it may obscure distinct relational patterns among these subcomponents. Although a longitudinal design was adopted, the study remains correlational in nature because it lacks experimental manipulation and therefore does not support strong causal inferences. The observed relationships may also be influenced by unmeasured variables, such as learning motivation, prior programming experience, or characteristics of the learning environment. Future research could incorporate experimental or quasi-experimental designs, together with more fine-grained dimensional analyses, to further clarify the causal mechanisms and internal structure linking AI literacy and computational thinking. Finally, although a longitudinal data collection procedure was employed, the two-month period may not be sufficient to capture the complex dynamics and long-term developmental trends in the relationships among these variables.

To address the limitations identified above, this study proposes the following directions for future research. First, to achieve a more comprehensive understanding of AI literacy, future research should incorporate multiple factors—such as the digital divide, perceived trust, and other relevant variables—into integrated analytical models. Multidimensional questionnaires could be developed to systematically examine the mechanisms through which these factors interact with AI literacy, as well as their differential effects across diverse demographic groups. Such an approach is expected to yield more holistic and nuanced insights. Additionally, greater attention should be devoted to analyzing cross-lagged relationships among relevant sub-dimensions (e.g., algorithmic thinking, critical thinking, ethical evaluation) and their interactions with contextual variables. Such fine-grained analyses will help elucidate the internal structure and dynamic linkages between computational thinking and AI literacy. Second, to enhance the representativeness and generalizability of the results, subsequent studies should expand the sample size, with particular emphasis on achieving a more balanced gender ratio (e.g., 1:1). By ensuring greater gender diversity within the sample, future research will be better positioned to accurately reflect the realities of AI literacy across different gender groups, thereby improving the applicability and reliability of the findings. Finally, to more effectively capture dynamic changes over time, future longitudinal designs should increase the frequency of data collection and extend the observation period to at least three months. This would allow for a more nuanced observation and analysis of how key variables evolve and interact over time, providing a stronger empirical foundation for understanding the developmental trajectories of AI literacy.

## Conclusion

This study primarily examined how computational thinking influences the development of AI literacy. To achieve this objective, a dynamic partial least squares structural equation model (PLS-SEM) was constructed using both cross-sectional and longitudinal data to analyze the relationship between computational thinking and AI literacy over time. The study systematically investigated the evolving relationship between these two constructs by tracking data across multiple time points. Data were collected at two time points, yielding 216 valid responses. The model was subsequently estimated and evaluated in SmartPLS 4.0 to ensure the accuracy and reliability of the results.

The findings indicate that computational thinking is a positive predictor of AI literacy. Moreover, students' computational thinking and AI literacy levels both improved over time. The core contribution of this research lies in integrating

computational thinking and AI literacy within a unified framework and examining their relationship using both cross-sectional and longitudinal data. From a practical standpoint, the study suggests that enhancing computational thinking may effectively promote AI literacy, offering valuable insights for educators. These results underscore the importance of incorporating computational thinking into instructional design to support students' AI literacy development.

## Supporting information

**S1 Data. Raw data: anonymized raw data for the two time points (T0 and T1) in this study.**
(PDF)

**S1 Table. Questionnaire: complete scales of computational thinking and artificial intelligence literacy used in the study.**
(DOCX)

## Author contributions

**Conceptualization:** Lehui Huang, Xueshi Wu.

**Data curation:** Qiuyu Li, Tinglan Huang.

**Formal analysis:** Qiuyu Li.

**Funding acquisition:** Qiuyu Li.

**Methodology:** Lehui Huang, Xueshi Wu.

**Writing – original draft:** Qiuyu Li.

**Writing – review & editing:** Qiuyu Li, Tinglan Huang.

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
