## [Decision Letter · Decision Letter 0]

27 Aug 2025

PONE-D-25-21913The Impact of University Students’ Computational Thinking on AI Literacy: A Longitudinal Study Based on SEM-PLSPLOS ONE

Dear Dr. Huang,

Thank you for submitting your manuscript to PLOS ONE. After careful consideration, we feel that it has merit but does not fully meet PLOS ONE’s publication criteria as it currently stands. Therefore, we invite you to submit a revised version of the manuscript that addresses the points raised during the review process.

We look forward to receiving your revised manuscript.

Kind regards,

Chengliang Wang

Academic Editor

PLOS ONE

Journal Requirements:

2. We are unable to open your Supporting Information file [Supporting information.rar]. Please kindly revise as necessary and re-upload.

Reviewers' comments:

Reviewer's Responses to Questions

**Comments to the Author**

1. Is the manuscript technically sound, and do the data support the conclusions?

Reviewer #1: Yes

Reviewer #2: Yes

2. Has the statistical analysis been performed appropriately and rigorously? 

Reviewer #1: Yes

Reviewer #2: Yes

3. Have the authors made all data underlying the findings in their manuscript fully available?

Reviewer #1: Yes

Reviewer #2: Yes

4. Is the manuscript presented in an intelligible fashion and written in standard English?

Reviewer #1: Yes

Reviewer #2: Yes

5. Review Comments to the Author

Reviewer #1: Dear Authors,

Your study adopted a cross-lagged SEM framework to explore the relationship between CT and AIL, which is really meaningful and well-reported. I recommend strengthening the following aspects before your study is published.

1. Conceptual introduction: The core variables should be introduced in the first chapter, including at least their brief/plain-language definitions, why they are important, and how they can be connected with each other.

2. Rewrite the literature review to emphasize research gaps: The current chapter of the literature review is a series of documented previous studies, with very clear topic sentences. The problem is that the authors overemphasized what is known to us. I believe you should analyze and summarize the research gaps that drive you to your research questions and aims.

3. Could you please add theoretical rationales for CT to predict next-time AIL, and not in the reverse direction? E.g., AIL may be promoted through specific training and technology use, which should subsequently benefit CT skills. Then, why is your hypothesized direction conceptually sound?

4. Please provide references for your data cleaning process. E.g., https://doi.org/10.1007/s40299-021-00570-0

5. In the research model, a number of potential moderators are not mentioned, e.g., AI use experience, academic achievements, and gender. Is this gender bias common, and why? How much do you estimate it should influence or bias your results? You may check similar research findings and contexts like https://doi.org/10.1186/s41239-023-00403-8 Potential moderators should be discussed in your implications and limitations.

6. Conceptual synthesis: We currently know that AIL and CT are compound concepts, meaning that they have several subdimensions. However, they are treated as high-level variables, ignoring the specific aspects. This is also a limitation of your study. E.g., https://doi.org/10.1080/0144929X.2022.2072768

Overall, your study is a good one that has been well conducted and reported. Please kindly strengthen the above aspects, add high-quality impactful literature related to your topics, and conduct overall language and format checks. I will be glad to hear back from you.

Reviewer #2: This study presents a commendable longitudinal investigation examining the impact of computational thinking on AI literacy among 216 undergraduate students, though certain methodological limitations warrant attention.

The following points require strengthening:

1. Temporal Design Validity: The two-month interval between data collection points appears insufficient for robust longitudinal analysis. As the authors acknowledge, "two months may not be sufficient to capture the complex dynamics and long-term developmental trends in the relationships among variables." However, no empirical justification is provided for this timeframe choice. Evidence demonstrating that this duration is adequate to avoid test-retest effects and capture meaningful developmental changes in both computational thinking and AI literacy is needed.

2. Gender Distribution Imbalance: The sample exhibits a significant gender imbalance with a male-to-female ratio of approximately 2:3 (39% male, 61% female). As noted by the authors, this "may restrict the generalizability of the results to broader populations." Gender-specific analyses should be conducted to ensure no systematic gender differences affect the structural relationships between computational thinking and AI literacy.

3. AI Literacy Measurement Transparency: Given that AI literacy concepts are rapidly evolving, full disclosure of the 38-item scale adapted from Laupichler, Aster (29) in the supplementary materials is essential for replication and validity assessment.

4. Cross-sectional vs. Longitudinal Findings Discrepancy: The rejection of H3 (β = 0.021, p = 0.708) presents a critical contradiction requiring deeper exploration. While cross-sectional analyses demonstrate that "computational thinking significantly and positively predicts AI literacy" (H1: β = 0.507, H2: β = 0.567, both p < 0.001), the longitudinal predictive relationship from T0 computational thinking to T1 AI literacy is non-significant. This discrepancy undermines the core premise that computational thinking serves as a longitudinal predictor of AI literacy development. The authors must provide theoretical explanations for why cross-sectional associations do not translate to longitudinal predictions, particularly when both constructs show significant improvement over time (H4 and H5 supported).

5. Causal Inference Limitations: This observational study design precludes strong causal claims. The limitations section should explicitly state that causal relationships cannot be definitively established without experimental manipulation. Additionally, the exclusive focus on computational thinking as a predictor overlooks other potentially important variables mentioned in the literature, including "digital divide, perceived trust," learning environment, individual motivation, and prior programming experience.

6. Attrition Analysis: The study experienced substantial participant attrition from the initial sample of 301 students to the final 216 participants (28.2% attrition rate). Specifically, 39 participants from the initial sample did not complete the second-round questionnaire, and an additional 35 responses were excluded due to invalid completion patterns. However, no systematic analysis was conducted to examine whether participants who dropped out differed systematically from those who remained in terms of baseline computational thinking abilities, AI literacy levels, or demographic characteristics. This lack of attrition analysis limits the ability to assess potential selection bias and threatens the internal validity of the longitudinal findings. A comparison between completers and non-completers on key study variables should be provided to ensure that the observed relationships are not artifacts of differential dropout patterns.

6. PLOS authors have the option to publish the peer review history of their article (what does this mean?). If published, this will include your full peer review and any attached files.

Reviewer #1: No

Reviewer #2: No

---

## [Author Response · Author response to Decision Letter 1]

16 Sep 2025

Dear Dr Wang,

We are writing to resubmit our revised manuscript entitled " The Impact of University Students’ Computational Thinking on AI Literacy: A Longitudinal Study Based on SEM-PLS (Manuscript ID: PONE-D-25-21913)" for further consideration at PLOS ONE. We wish to express our deepest gratitude to you and the reviewers for the generous time and exceptional effort you have dedicated to reviewing our work. Your insightful and constructive feedback has been invaluable in guiding our revisions and significantly improving the quality of our manuscript.

We have given careful consideration to every comment and suggestion provided. In response, we have thoroughly revised the manuscript to address all the points raised. The modifications are clearly highlighted in the "Revised Manuscript with Track Changes" file for easy reference. Additionally, we have prepared a detailed point-by-point response to each comment, which is included in the "Response to Reviewers" document.

We sincerely appreciate the opportunity to revise our manuscript. The feedback has not only strengthened this paper but has also provided us with important guidance for our future research. We are very grateful for the supportive and rigorous review process.

Thank you once again for your time and consideration. We are deeply appreciative of the work that you and the reviewers have done. We hope that the revised manuscript now meets the high standards of PLOS ONE and look forward to hearing from you.

With warmest thanks,

Lehui Huang

Jiangxi Science and Technology Normal University

Email: hlh8899@jxstnu.edu.cn

---

## [Decision Letter · Decision Letter 1]

14 Nov 2025

PONE-D-25-21913R1The Impact of University Students’ Computational Thinking on AI Literacy: A Longitudinal Study Based on SEM-PLSPLOS ONE

Dear Dr. Huang,

Thank you for submitting your manuscript to PLOS ONE. After careful consideration, we feel that it has merit but does not fully meet PLOS ONE’s publication criteria as it currently stands. Therefore, we invite you to submit a revised version of the manuscript that addresses the points raised during the review process.

Please note Reviewer #2's outstanding concerns.

We look forward to receiving your revised manuscript.

Kind regards,

Avanti Dey, PhD

Staff Editor

PLOS ONE

Journal Requirements:

Reviewers' comments:

Reviewer's Responses to Questions

**Comments to the Author**

1. If the authors have adequately addressed your comments raised in a previous round of review and you feel that this manuscript is now acceptable for publication, you may indicate that here to bypass the “Comments to the Author” section, enter your conflict of interest statement in the “Confidential to Editor” section, and submit your "Accept" recommendation.

Reviewer #2: All comments have been addressed

2. Is the manuscript technically sound, and do the data support the conclusions?

Reviewer #2: Partly

3. Has the statistical analysis been performed appropriately and rigorously? 

Reviewer #2: Yes

4. Have the authors made all data underlying the findings in their manuscript fully available?

Reviewer #2: Yes

5. Is the manuscript presented in an intelligible fashion and written in standard English?

Reviewer #2: Yes

6. Review Comments to the Author

Reviewer #2: 1) There are quite a few typos.

2) Also, while you're performing a t-test, this requires normality, so please include the results of normality tests.

3) This is the main revision. In the abstract, it states “novel contribution...establishing a longitudinal link,” but the longitudinal hypothesis H3 (CT at T0 → AIL at T1) is not significant (β=0.021, p=0.708). Therefore, this claim of novelty is inconsistent with the current results. Consequently, the claim does not hold. Please address this inconsistency.

7. PLOS authors have the option to publish the peer review history of their article (what does this mean?). If published, this will include your full peer review and any attached files.

Reviewer #2: No

---

## [Author Response · Author response to Decision Letter 2]

18 Nov 2025

Subject: Rebuttal Letter: Manuscript PONE-D-25-21913R1

Dear Editor and Reviewers,

Thank you for granting us the opportunity to revise our manuscript titled “The Impact of College Students' Computational Thinking on Artificial Intelligence Literacy: A Longitudinal Study Based on SEM-PLS” (Manuscript ID: PONE-D-25-21913R1). We sincerely appreciate the valuable time and effort you and the reviewers have invested in providing insightful and constructive feedback on our work.

We have carefully reviewed all comments and made comprehensive revisions to the manuscript. We believe these modifications have significantly enhanced the quality and clarity of the paper.

Below is our point-by-point response to each comment raised by the reviewers. All modifications in the manuscript are highlighted in red for your reference.

Responses to Reviewer2

(The reviewer's original comments are displayed in blue, while the revised responses are highlighted in red.)

Reviewer #2

Comment 1:

There are quite a few typos.

Response 1:

We appreciate the reviewers' valuable feedback on language expression. Accordingly, we have proofread and polished the entire manuscript to enhance readability by correcting spelling and grammatical errors. We hope the revised version is now clearer and more accessible for readers to understand.

Comment 2:

Also, while you're performing a t-test, this requires normality, so please include the results of normality tests.

Response 2:

We agree with the reviewer that this is an important methodological point. As suggested, we performed the Shapiro-Wilk test to assess the normality of the baseline continuous variables. The results indicated that the distributions for the CT and AIL scores significantly deviated from normality (p < 0.05).

Therefore, to ensure the robustness of our findings, we additionally conducted nonparametric Mann-Whitney U tests for verification. The results from these nonparametric tests were fully consistent with the original t-test findings:

CT score: U = 8082.500, p = 0.105

AIL score: U = 8517.500, p = 0.329

We have now integrated both the normality test results and the results of the Mann-Whitney U tests into the revised manuscript. These revisions are located between lines 275 and 278 on page 13.

This addition strengthens our methodological rigor and confirms that our initial conclusions are valid irrespective of the parametric assumptions.

Comment 3:

This is the main revision. In the abstract, it states “novel contribution...establishing a longitudinal link,” but the longitudinal hypothesis H3 (CT at T0 → AIL at T1) is not significant (β=0.021, p=0.708). Therefore, this claim of novelty is inconsistent with the current results. Consequently, the claim does not hold. Please address this inconsistency.

Response 3:

We greatly appreciate the reviewer's critical and insightful comment. The reviewer's perspective is entirely correct. We acknowledge that the original assertion of “establishing a longitudinal association” was overly assertive and not supported by the non-significant results of H3.

To address this issue, we have taken the following actions:

We have revised the statement in the abstract. The original sentence has been modified to: “This study employs a longitudinal research approach to elucidate the complex relationship between computational thinking and AI literacy, providing theoretical insights and methodological guidance for educational practice, thereby contributing to the literature in this field.” These revisions are located between lines 39 and 42 on page 2. The new phrasing more accurately reflects the exploratory longitudinal design of this study and avoids overstating the research findings.

In the discussion section (lines 425–457, pages 20–22), we explicitly explain the reasons for the lack of significance in the longitudinal study (H3 hypothesis), proposing two core interpretations:

Multifactorial nature of learning: We note that in real educational settings, the development of AI literacy is influenced by multiple factors (e.g., curriculum content, peer interactions, teacher guidance), which may have weakened the independent predictive power of baseline computational thinking during the intervention period.

Methodological considerations: We analyzed the potential impact of ceiling effects in short-term longitudinal designs and the sensitivity of AI literacy measurement tools.

We believe these revisions not only resolve inconsistencies in the findings but also enhance the study's academic contribution by deepening our understanding of the relationship between critical thinking and autonomous learning.

---

## [Decision Letter · Decision Letter 2]

5 Apr 2026

PONE-D-25-21913R2The Impact of University Students’ Computational Thinking on AI Literacy: A Longitudinal Study Based on SEM-PLSPLOS One

Dear Dr. Huang,

Thank you for submitting your manuscript to PLOS ONE. After careful consideration, we feel that it has merit but does not fully meet PLOS ONE’s publication criteria as it currently stands. Therefore, we invite you to submit a revised version of the manuscript that addresses the points raised during the review process.

We look forward to receiving your revised manuscript.

Kind regards,

Dr. Manuel Salas-Velasco, PhD

Academic Editor

PLOS One

Journal Requirements:

**Additional Editor Comments:**

Overall, the reviewers agree that the manuscript is technically sound, the statistical analyses are appropriately conducted, and the data adequately support the conclusions. The revisions made in response to previous reviewer comments have substantially improved the clarity, rigor, and balance of the manuscript. In particular, the adjustment of claims regarding longitudinal effects and the strengthened methodological transparency are noted positively.

Reviewer 3 raises several points that are largely editorial or contextual in nature (e.g., style of the abstract, language polishing, discussion of generalizability, and broader contextualization of AI-related risks). These suggestions are reasonable and can be addressed through minor revisions aimed at improving clarity and discussion, without requiring any changes to the study design, analyses, or results.

One comment related to the justification of AMOS and covariance-based SEM (CB-SEM) is not applicable, as the manuscript clearly employs PLS-SEM using SmartPLS, with an appropriate methodological justification already provided. This point does not require further action.

Accordingly, the editorial assessment is that no major methodological or conceptual concerns remain. The manuscript would benefit from a minor revision focused on:

•  Light stylistic refinement of the abstract (if desired),

•  Minor language polishing,

•  Briefly reinforcing the discussion of sample size, timeliness, and broader educational implications of AI, including potential risks, at a conceptual level.

Reviewers' comments:

Reviewer's Responses to Questions

**Comments to the Author**

1. If the authors have adequately addressed your comments raised in a previous round of review and you feel that this manuscript is now acceptable for publication, you may indicate that here to bypass the “Comments to the Author” section, enter your conflict of interest statement in the “Confidential to Editor” section, and submit your "Accept" recommendation.

Reviewer #2: All comments have been addressed

Reviewer #3: All comments have been addressed

2. Is the manuscript technically sound, and do the data support the conclusions?

Reviewer #2: Yes

Reviewer #3: Yes

3. Has the statistical analysis been performed appropriately and rigorously? 

Reviewer #2: Yes

Reviewer #3: Yes

4. Have the authors made all data underlying the findings in their manuscript fully available?

Reviewer #2: Yes

Reviewer #3: Yes

5. Is the manuscript presented in an intelligible fashion and written in standard English?

Reviewer #2: Yes

Reviewer #3: Yes

6. Review Comments to the Author

Reviewer #2: Regarding the three comments, they are partially unmet.

Comment 1:

While not a major issue, please perform careful proofreading once more before final acceptance. For example:

・In Reference 5, the space is missing between “Framework Construction and Enhancement Paths” and Construction and.

・In the conference name of the reference, “WIPSCE 20242024” has duplicate years.

・In the labels of Figure 2 (Structural Model), “T0-Computation thinking”, “T1- Computation thinking” are missing the letter ‘l’ in “Computational”.

Comment 2:

The Shapiro–Wilk test statistic and p-value itself are not reported in the main text. The paragraph above only states “baseline continuous variables might deviate from a normal distribution” without reporting any normality test values.

Furthermore, while the authors emphasize paired t-tests for T0–T1 changes (H4, H5) in this paper, that section (Table 7 and surrounding text) does not include any mention of normality or results from nonparametric tests like the Wilcoxon signed-rank test.

Comment 3:

Regarding the phrasing “positive predictor” in the conclusion, it would be safer to soften it to explicitly indicate a cross-sectional prediction (association at the same time point), such as “cross-sectional models showed that CT is a positive predictor of AIL at each time point.”

Reviewer #3: Dear Editor, Dear Authors,

I can see that the authors have revised the manuscript carefully, and I believe the paper can be accepted after minor revisions. Please consider the following comments before final acceptance.

Abstract: Please remove numerical values from the abstract, as numbers alone do not communicate the study’s meaning clearly. Instead, the abstract should briefly state the background, purpose, key findings, and especially the implications in a more descriptive way.

Language quality: There are indications that AI tools may have been used in drafting the manuscript. The paper should undergo careful human proofreading (or professional language editing) to improve clarity, coherence, and academic tone before acceptance.

Timeliness of the study: The study appears to be based on data collected in 2024. Given that it is now 2026 and the AI field evolves rapidly, the authors should justify the continued relevance of the findings and/or explicitly acknowledge this as a limitation.

Sample size and potential bias: The sample size is relatively small. Please explain how the authors ensured the findings are not strongly affected by sampling bias, and discuss how the sample size may influence generalizability. If applicable, add a limitation and suggestions for future research with larger samples.

Methodology (AMOS and CB-SEM justification): The authors should clearly justify why AMOS was used and briefly explain what covariance-based SEM (CB-SEM) is, including why it is appropriate for this study. A recent example with clear methodological explanation is:

(2024). Exploring the factors affecting elementary mathematics teachers’ innovative behavior: An integration of social cognitive theory. Scientific Reports, 14(1), 1–14. https://doi.org/10.1038/s41598-024-52604-4

Discussion (negative effects of AI): The discussion should also acknowledge and analyze potential negative effects or risks of AI in education (e.g., overreliance, AI dependency, bias, reduced critical thinking, ethical concerns). The discussion can be strengthened by engaging with recent related literature, such as:

(2024). Examining Chinese preservice mathematics teachers’ adoption of AI chatbots for learning: Unpacking perspectives through the UTAUT2 model. Education and Information Technologies.

(2024). Latent profile analysis of AI literacy and trust in mathematics teachers and their relations with AI dependency and 21st-century skills. Behavioral Sciences (Basel, Switzerland), 14.

(2025). Exploring the relationship between AI literacy, AI trust, AI dependency, and 21st-century skills in preservice mathematics teachers. Scientific Reports, 15(1).

7. PLOS authors have the option to publish the peer review history of their article (what does this mean?). If published, this will include your full peer review and any attached files.

Reviewer #2: No

Reviewer #3: No

---

## [Author Response · Author response to Decision Letter 3]

7 May 2026

Editor

Comment 1:

Light stylistic refinement of the abstract (if desired).

Response 1:

Thank you for this suggestion. We have made an effort to refine the abstract in terms of style and conciseness while trying to preserve the core findings and meaning. The language has been lightly polished with the aim of improving its flow and readability.

Comment 2:

Minor language polishing.

Response 2:

Thank you for this recommendation. We have carefully gone through the manuscript and made minor language revisions throughout the text. We have attempted to improve sentence structure, word choice, and grammatical accuracy to enhance overall clarity.

Comment 3:

Briefly reinforcing the discussion of sample size, timeliness, and broader educational implications of AI, including potential risks, at a conceptual level.

Response 3:

Thank you for this insightful suggestion. In response, we have made two targeted additions to the manuscript.

First, to address sample size and timeliness, we have added the following text in the Limitations section (see page 27, lines 566–584).

Second, to reinforce the broader educational implications and potential risks of AI, we have added the following text in the Discussion section (see page 23, lines 474–489).

We hope these additions help to address the conceptual points raised and contribute to a more balanced discussion.

---

## [Editor Report · Decision Letter 3]

11 May 2026

The Impact of University Students’ Computational Thinking on AI Literacy: A Longitudinal Study Based on SEM-PLS

PONE-D-25-21913R3

Dear Dr. Huang,

We’re pleased to inform you that your manuscript has been judged scientifically suitable for publication and will be formally accepted for publication once it meets all outstanding technical requirements.

Kind regards,

Dr. Manuel Salas-Velasco, PhD

Academic Editor

PLOS One
---

## [Editor Report · Acceptance letter]

PONE-D-25-21913R3

PLOS One

Dear Dr. Huang,

I'm pleased to inform you that your manuscript has been deemed suitable for publication in PLOS One. Congratulations! Your manuscript is now being handed over to our production team.

Kind regards,

on behalf of

Dr. Manuel Salas-Velasco

Academic Editor

PLOS One